



# Five satellite sensor study of the rapid decline of wildfire smoke in the stratosphere

Bengt G. Martinsson*, Johan Friberg, Oscar S. Sandvik, and Moa K. Sporre
Department of Physics, Lund University, Lund, Sweden

*Email: bengt.martinsson@nuclear.lu.se



## Abstract

Smoke from Western North American wildfires reached the stratosphere in large amounts in
August 2017. Limb-oriented satellite-based sensors are commonly used for studies of wildfire
aerosol injected into the stratosphere (OMPS-LP (Ozone Mapping and Profiler Suite Limb
Profiler) and SAGE III/ISS (Stratospheric Aerosol and Gas Experiment III on the International
Space Station)). We find that these methods are inadequate for studies the first 1 – 2 months after
such a strong fire event due to event termination ("saturation"). The nadir-viewing lidar CALIOP
(Cloud-Aerosol Lidar with Orthogonal Polarization) is less affected due to shorter path in the
smoke, and, further, provides means that we could use to develop a method to correct for strong
attenuation of the signal. After the initial phase, the aerosol optical depth (AOD) from OMPS-LP
and CALOP show very good agreement above the 380 K isentrope, whereas the OMPS-LP tends
to produce higher AOD than CALIOP in the lowermost stratosphere (LMS), probably due to
reduced sensitivity at altitudes below 17 km. Time series from CALIOP of attenuation-corrected
stratospheric AOD of wildfire smoke show an exponential decline during the first month after the
fire, which coincides with highly significant changes in the wildfire aerosol optical properties.
The AOD decline is verified by the evolution of the smoke layer composition, comparing the
aerosol scattering ratio (CALIOP) to the water vapor concentration from MLS (Microwave Limb
Sounder). Initially the stratospheric wildfire smoke AOD is comparable with the most important
volcanic eruptions during the last 25 years. Wildfire aerosol declines much faster, 80 – 90% of
the AOD is removed with a half-life of approximately 10 days. We hypothesize that this dramatic
decline is caused by photolytic loss. This process is rarely observed in the atmosphere. However,
in the stratosphere this process can be studied with practically no influence from wet deposition,
in contrast to the troposphere where this is the main removal path of sub-micron aerosol particles.
Despite the loss, the aerosol particles from wildfire smoke in the stratosphere are relevant for the
climate.

## 1. Introduction

Background stratospheric aerosol is composed of sulfuric acid, water, carbonaceous components,
and minor extraterrestrial and tropospheric components (Murphy et al., 2007; Kremser et al.,
2016; Martinsson et al., 2019). Volcanism is a strong source of the stratospheric sulfurous,
carbonaceous and ash aerosol (Martinsson et al., 2009; Andersson et al., 2013; Friberg et al.,
2014). Large eruptions, like that of Mt Pinatubo in 1991, affect the stratosphere for several years,
causing global cooling of several tenths of degrees Kelvin (Kremser et al., 2016). These eruptions
are scarce, only a few per century (Ammann et al., 2003; Stothers, 2007). Moderate eruptions are
more frequent contributors to the stratospheric aerosol (Vernier et al., 2011; Andersson et al.,
2015; Friberg et al., 2018), forming the persistently variable stratospheric background aerosol
(Solomon et al., 2011).

The stratospheric aerosol is also influenced by pyrocumulonimbus clouds (pyroCb) that form
during extreme weather conditions in connection with intense wildfires (Fromm et al., 2010). The



ongoing climate change is projected to increase the frequency of large wildfires (Kasischke et al.,
2006; Dennison et al., 2014). Interestingly, the two largest events have, in terms of stratospheric
impact, occurred during the last few years, in North America 2017 (Peterson et al., 2018) and
Australia 2019-2020 (Kablick et al., 2020). Here we investigate the great pyroCbs formed in
western North America on August 12 – 13, 2017. Figure 1a shows an example on the strong
impact on the stratospheric aerosol of the 2019 Raikoke volcanic eruption, one of the strongest
eruptions post Mt Pinatubo in 1991. In comparison, Figure 1b demonstrates the formidable early
impact of wildfire aerosol. The stratospheric impact of that fire has been described in terms of
light-backscatter reaching unprecedentedly high values for a non-volcanic aerosol layer (Khaykin
et al., 2018), light extinction about 20 times higher than after the Pinatubo volcanic eruption in
1991 (Ansmann et al., 2018), and mass of smoke comparable to that of a moderate sized volcanic
eruption (Peterson et al., 2018). The pyroCbs lifted smoke from the fire to the extratropical
tropopause region, where absorption of radiation by black carbon (BC) in the smoke induced
additional lift to 23 km altitude in 2 months (Yu et al., 2019).
Smoke particles from wildfires contain a dominating fraction of organic matter by mass
(Garofalo et al., 2019). Organic aerosol is susceptible to photochemical loss (Jimenez et al.,
2009), and laboratory studies have demonstrated that this phenomenon could be an important
sink of secondary aerosol mass (Molina et al., 2004; Sareen et al., 2013). The residence time of
stratospheric air spans months to years depending on its path in the Brewer-Dobson circulation
(Engel et al., 2009; Bönisch et al., 2009). Due to very low probability of clouds, fine aerosol
particles have considerably longer residence times in the stratosphere than in the troposphere,
which further emphasizes the importance of investigating photochemical loss in the stratosphere
(Martinsson et al., 2019).
The aim of this study is to further understand the stratospheric aerosol sources and its climate
impact. We develop methodology to correct for attenuation in dense smoke layers from wildfires
to properly deal with intense smoke injections into the stratosphere, with two main questions: 1)
does photochemical loss of wildfire smoke occur in the stratosphere, and 2) how does the AOD
of smoke from the wildfire studied here compare with volcanic aerosol?
The first decade of the 21st century was characterized by slower temperature evolution than
anticipated from CMIP5 models (Fyfe et al., 2016). The discrepancy was attributed to inter-
decadal Pacific oscillation (Medhaug et al., 2017), variations in solar forcing (Myhre et al., 2013)
and aerosol in the stratosphere from moderate volcanic eruptions (Santer et al., 2014). Should
wildfire smoke in the stratosphere be added to this list of phenomena that require more attention
in climate models?
Our investigation deals with the evolution of the wildfire AOD, and aerosol optical properties
obtained from the lidar CALIOP aboard the CALIPSO (Cloud-Aerosol Lidar and Infrared
Pathfinder Satellite Observation) satellite, OMPS-LP/Suomi  and SAGE III/ISS in comparison
with volcanic injections to the stratosphere. Additionally, the water vapor concentrations of
individual smoke layers are investigated by the MLS, the spatial evolution of smoke layers is





investigated using OMPS-NM (Ozone Mapping and Profiler Suite Nadir Mapper), and the AODs
and extinction coefficients obtained from CALIOP are compared with that of OMPS-LP and
SAGE III/ISS.
**2. Methods**
This study of the dense stratospheric smoke layers from pyro-cumulonimbus formed over
Western North America in August 12 – 13, 2017 is based on five satellite sensors. For four of
them, OMPS-LP, SAGE III/ISS, MLS and OMPS-NM, high level products (Level 2) are used.
The CALIOP data evaluation is based on a Level 1 product. A method to correct for attenuation
of the CALIOP laser beam in the smoke layers is presented. For these reasons CALIOP requires
more space in this section compared to the other methods.
*2.1 CALIOP*
The evaluation of the CALIOP instrument carried by the CALIPSO satellite is based on version
4-10, level 1B data. CALIOP measures backscattering of laser light at two wavelengths, 532 and
1064 nm. For the shorter wavelength, scattered laser light is detected in parallel and
perpendicular polarizations relative to the outgoing beam. These almost nadir-viewing aerosol
and cloud measurements result in high resolution vertical profiles. For the altitude ranges <8.2,
8.2 – 20.2, 20.2 – 30.1 and 30.1 – 40 km the vertical resolutions are 30, 60, 180, and 300 m,
respectively. CALIPSO orbits between 82° S and 82° N, completing 14 – 15 orbits per day
(Winker et al., 2007; Winker et al., 2010).
*2.1.1 AOD*
Stratospheric AOD was obtained by integrating the backscattering intensity corrected for
attenuation (described below) from the tropopause to 35 km altitude. Figure 1b illustrates how
attenuation of the laser signal strongly reduced the signal below the dense smoke layer between
11 to 16 km altitude. We use the tropopause height according to MERRA-2 supplied with the
version 4.10 CALIOP data, which is a mixture of a dynamic and a thermal tropopause. The AOD
was averaged in the 20 - 80° N latitude range, where all nighttime swaths available from
CALIOP were included. The data were averaged over all longitudes in one-degree latitude bands,
and these latitude bands were averaged for the 20 - 80° N latitude range using area-weighting.
For dense layers, the lidar ratios estimated for the individual smoke layers were applied
(explained below). Apart from the first few days the lidar ratio shows no temporal evolution, it is
found to have geometrical mean of 48.9 sr with double-sided 95% confidence interval of 47.6 –
50.3 sr (Figure 2a), which is close to the typical background lidar ratio of 50 sr (Jäger and
Deshler, 2003). For layers that were not dense, the lidar ratio was held at this typical background
level. The volume depolarization ratio ($\delta_v$) contains information that can be used to classify
aerosol layers. When the depolarization ratio is less than 0.05 the data is considered background
and the lidar ratio is set to 50 sr. Ice-clouds were removed in the lowest 3 km of the stratosphere
by identifying them in stratospheric layers where the backscattering was high (attenuated
backscattering larger than 0.0025 km$^{-1}$ sr$^{-1}$). Data in these layers were classified as probable
clouds if their $\delta_v$ was higher than 0.20, or smoke if $\delta_v$ was between 0.05-0.20, after which the data





within each swath were clustered depending on their location. Noise in the data led to some lone
pixels within layers of either ice or smoke. These were reclassified depending on the surrounding
pixels, making sure that no single pixel marked as aerosol occurred within the ice-cloud layers.
Layers of ice-clouds were then expanded upwards and horizontally to capture faint edges of the
clouds (Friberg et al., 2018). The classification was carried out on data at 8 km resolution along
each swath with their highest vertical resolution (30, 60, or 180 m, depending on altitude), after
which the tropospheric data were removed. Possible polar stratospheric cloud (PSC) signals north
of 45°N were excluded by classifying pixels with temperature below 195 K as possible PSC
occasions. Underlying pixels were also excluded, to prevent bias from attenuation of the lidar
signals or from settling ice-crystals (Friberg et al., 2018).

*2.1.2 Attenuation correction and radiative properties of individual smoke layers*

The evolution of the lidar, color and depolarization ratios were investigated using 32 separate
smoke layer measurements over the period 3 – 59 days after the fire. CALIOP has a statistical
disadvantage compared with lidars at the ground (Baars et al., 2019), because of small solid angle
due to long distance to the stratosphere (~700 km) and short measurement time. Optical
properties of old and faint individual smoke layers therefore could not be quantified with high
precision using CALIOP. The faint layers though still affect the AOD determinations described
above, where AOD elevation after the fire remains approximately one year. Out of the 32 smoke
layers studied, 29 were night-time measurements, whereas the remaining three are defined as
day-time measurements. These latter ones increased the number of early observations (day 3 – 5)
and were taken when the disturbance from solar radiation is small, i.e., shortly before the night.

The first weeks after the fire the smoke layers could be very dense with layer AODs exceeding 1,
causing strong attenuation of the CALIOP signals with two-way transmissions down to below
0.01. For the 532 nm wavelength the particle lidar ratio was estimated by aiming the scattering
ratio (R; total-to-molecular backscattering ratio) below a smoke layer to a target value. The target
value was obtained from the background scattering ratio beside each smoke layer investigated,
which on average is R = 1.08, with standard deviation ±0.05. To reduce influence from noise, the
CALIOP data were averaged along the swath. The averaging range varied between the smoke
layers, due to its extension along the swath, the homogeneity of the layer, and avoidance of sub-
layer features.

The particle lidar ratio of an individual smoke layer was iterated until reaching the target value (R
= 1.08) described above from the combined effect of all altitude pixels. Pixels at altitudes outside
the smoke layer were set to the background lidar ratio of 50 sr (Jäger and Deshler, 2003). The
altitude resolution provided in the CALIOP data was used, where each altitude pixel (j) is
corrected for attenuation. The calculation starts at the highest altitude (40 km) and continues
downwards in two rounds. In the first round the star-marked quantities of equations 1-3 were
computed, correcting for attenuation from overlaying pixels. Before moving to the next altitude,
we account for self-attenuation from the pixel itself (equations to the right, without a star):



$$\beta_j^* = \frac{\beta_j'}{\prod_{k=1}^{j-1} T_k^2}; \qquad \beta_j = \frac{\beta_j^*}{\sqrt{T_j^{*2}}} \qquad (1)$$


where $\beta'$ is the attenuated backscattering and $T^2$ the two-way transmissions from both particles
and molecules. The two-way particle transmission is obtained by first computing the AOD:

$$AOD_j^* = \left(\beta_j^* - \beta_{m,j}\right) S_p \Delta z_j; \qquad AOD_j = \left(\beta_j - \beta_{m,j}\right) S_p \Delta z_j \qquad (2)$$


where $\Delta z_j$ is the height of the altitude pixel, $\beta_{m,j}$ is backscattering from air molecules, and $S_p$ the
lidar ratio of the aerosol particles. The molecular lidar ratio, for computation of the molecular
extinction, was set to 8.70477 sr (Prata et al., 2017). CALIOP measurements are affected by
multiple scattering (Wandinger et al., 2010), causing overestimation of the backscattering. The
multiple scattering factor ($\eta$), the ratio of the apparent to the actual extinction coefficient, is not
known. Previous estimates are in the range $0.85 - 0.95$ for layers thicker than 500 m (Prata et al.,
2017). Not correcting for multiple scattering results in determination of the effective lidar ratio,
which is lower than the actual lidar ratio by a factor $\eta$. In equation 2 thus the backscattering
inflated by multiple scattering is multiplied by an underestimated lidar ratio to, at least in part,
compensate for the effects of multiple scattering on the AOD. The two-way transmission of
altitude pixel j due to the particles present is obtained from:

$$T_{p,j}^{*2} = \exp\left(-2AOD_j^*\right); \qquad T_{p,j}^2 = \exp\left(-2AOD_j\right) \qquad (3)$$


These calculations in equations $1 - 3$ are carried out until the background layer between altitudes
a and b below the smoke layer reaches the target scattering ratio of 1.08 (Figure 3a):

$$R = \frac{\sum_a^b \beta_j}{\sum_a^b \beta_{m,j}} \qquad (4)$$


Error estimates of the effective lidar ratio were obtained by varying the target scattering ratio
from its average value (R = 1.08) mentioned above, to its ±0.05 standard deviation range. The
fitting uncertainty in these estimates is strongly dependent on the light extinction in the smoke
layer. Dense layers result in very small uncertainties in the effective lidar ratio because of the
strong impact on R from a slight change in the extinction. Layers with lower extinction
progressively increase the uncertainties of the estimate. When the error estimate of the effective
lidar ratio fit exceeds 25% the result is excluded from the data analysis, which terminates
estimates of lidar ratios from day 22 after the fire.

The color ratio, the ratio between the backscattering at 1064 nm to 532 nm wavelength, is
affected by a difference in attenuation of the two wavelengths. This is clearly visible for dense
smoke layers in the CALIOP browse images by a gradual increase of the color ratio through the
layer because of the weaker attenuation for 1064 nm wavelength than for 532 nm (Figure 1d).



Therefore, estimations of the attenuation were undertaken also for the long wavelength. The
molecular backscattering is assumed to be 1/16 of that at 532 nm ($1/\lambda^4$ dependence of Rayleigh
scattering). Weak molecular scattering at 1064 nm prohibits lidar ratio estimation at that
wavelength by CALIOP. Instead, the lidar ratio was assumed to be 60 sr, inducing uncertainties
in the color ratio. The volume color ratio is obtained from:
$$\chi = \sum_{k=top}^{base} \beta_{1064,k} / \sum_{k=top}^{base} \beta_{532,k} \qquad (5)$$
To limit influence from attenuation in the color ratio computations, the estimates were based on
the upper part of a smoke layer. Starting from the top of the smoke layer, the computations were
truncated when the two-way transmission of the 532 nm wavelength fell below 0.7. Varying the
1064 nm wavelength lidar ratio in the wide range of 60 ±20 sr the uncertainty in the color ratio
becomes less than ±5% with this constraint applied. From the color ratio we define the particle
color ratio:
$$\chi_p = \sum_{k=top}^{base}(\beta_{1064,k} - \beta_{m,1064,k}) / \sum_{k=top}^{base}(\beta_{532,k} - \beta_{m,532,k}) = \frac{\chi R}{R-1} - \frac{1}{16(R-1)} \qquad (6)$$
where we made use of the wavelength dependence of Rayleigh scattering for molecular
scattering, and the scattering ratio for the 532 nm wavelength was obtained from eqn. 4.
We also investigated the depolarization of the scattered laser beam at 532 nm by first forming the
volume depolarization ratio:
$$\delta_v = \sum_{k=top}^{base} \beta'_{532\perp,k} / \sum_{k=top}^{base} \beta'_{532,k} \qquad (7)$$
where symbol $\perp$ indicates scattered light polarized perpendicularly to the incident beam. Having
access to the volume depolarization and an estimate of the molecular depolarization ratio $\delta_m \approx$
0.003656 (Prata et al., 2017; Hostetler et al., 2006) the particle depolarization ratio is obtained
from:
$$\delta_p = \frac{\delta_v - \delta_m + \delta_v(1+\delta_m)(R-1)}{\delta_m - \delta_v + (1+\delta_m)(R-1)} \qquad (8)$$
where R is obtained from eqn. 4.
*2.2 Extinction coefficients and AOD from OMPS-LP*
The aerosol data from OMPS-LP (Chen et al., 2018; Jaross et al., 2014; Loughman et al., 2018)
have lately been used extensively in the literature on volcanic and wildfire impact on the
stratospheric aerosol. Several data products are available, here we use the recently released Level
2 product: Suomi-NPP OMPS LP L2 AER Daily Product, version 2.0 (Taha et al., 2020). The
polar-orbiting Suomi satellite completes 14 - 15 laps per day. OMPS-LP is a limb-scattering



method that collects data looking backwards along the satellite orbit, and along two other
directions separated by 4.25° from the orbit, giving a cross-track separation of approximately 250
km at the tangent point. Measurements are undertaken in the wavelength and altitude ranges of
290 – 1000 nm and 10 – 80 km, respectively. The vertical resolution of OMPS-LP is 1.5 – 2 km
(Rault and Loughman, 2013). The aerosol product used here comprises 6 wavelengths (510, 500,
675, 745, 869 and 997 nm). The group responsible for the OMPS-LP version 2.0 data (Taha et
al., 2020) recommends caution when using data from altitudes below 17 km altitude due to loss
of sensitivity. This problem can be reduced by use of the 745 nm and longer wavelengths. Here
we will make use of two of wavelengths: 745 nm because of the reduced problem with
sensitivity, and 510 nm because it is the wavelength closest to that of CALIOP (532 nm).
The OMPS-LP aerosol extinction coefficients are provided on a grid with a vertical resolution of
1 km. To study the smoke from the August 2017 fire we compute the average AOD over all
longitudes in the latitude interval 20 – 80° N for three layers, the LMS (tropopause to 380 K
isentrope), lower Brewer-Dobson branch (380 – 470 K) and the upper Brewer-Dobson branch
(470 K to 35 km altitude). The OMPS-LP version 2 dataset use a cloud detection algorithm (Chen
et al., 2016), and comes in two forms: one without filtering out signals from clouds, and the other
where signals affected by clouds and polar stratospheric clouds are removed. In Figure 4 we
show both these varieties for 745 nm wavelength, and, with and without flags regarding data
quality including profile retrieval errors (named RetrievalFlags in the OMPS-LP files), high root-
mean squares (ResidualFlags), and further errors from the South Atlantic anomaly, disturbances
from the Moon, solar eclipses, planets, and satellite maneuvers (SwathLevelQualityFlags). In the
two upper layers (Figures 4a and b) the differences are usually small between the varieties except
for some spikes, whereas the LMS data (Figure 4c) show large stochastic variability as well as
periods of clear differences between the varieties. Since this data is taken well below 17 km
altitude, sensitivity issues can be expected (Taha et al., 2020), see above. Days 130 – 190 (during
December 2017 to February 2018) several spikes appear in the two higher layers which likely are
caused by polar stratospheric clouds. The data set filtered for clouds and flagged stands out by
comparably small peaks, whereas the differences between the varieties usually are small
elsewhere. We therefore select the cloud-filtered and flagged data for further analysis in the
coming sections.
*2.3 Extinction coefficients from SAGE III/ISS*
SAGE III/ISS is a limb-viewing instrument based on solar occultation. Here we make use of
Level 2 aerosol extinction coefficients (SAGE III/ISS User's Guide, 2018), version 5.10,
supplied with a vertical resolution of 0.5 km. The upper limit of the slant path optical depth is
about 8, translating to a vertical optical depth of approximately 0.02 (SAGE III/ISS User's Guide,
2018). The orbiting of ISS differs markedly from the polar orbiting satellites CALIPSO
(CALIOP) and Soumi (OMPS-LP). This causes sporadic coverage by ISS of the latitudes of
interest here, resulting in that no average AODs over the 20 – 80° N latitude range could be
formed with adequate time resolution. However, daily maximum extinction coefficients from
SAGE III/ISS could, when available, be included in a comparison with CALIOP and OMPS-LP.




### *2.4 Water vapor measurements from MLS*


Water vapor concentrations (mixing ratio) in individual smoke layers was obtained from the MLS
instrument aboard the Aura satellite (Waters et al., 2006) in 12 vertical steps per decade of
pressure (version 5.0-1.0a, level 2). In nighttime measurements from days 6 – 59 after the fire,
the smoke layers studied by CALIOP were also investigated with MLS in almost simultaneous
measurements, both instruments being on satellites that are members of the A-train. Data in the
10 – 316 hPa atmospheric pressure range were used, with vertical resolution 1.3 – 3.2 km
(Livesley et al., 2020). Limited vertical resolution induces problems to obtain well defined
observation of $H_2O$ concentration of smoke layers close to the strong $H_2O$ concentration gradient
across the tropopause. Close to the tropopause, but in the stratosphere, no $H_2O$ peak from a
smoke layer can be detected. As the distance to the tropopause increases, an $H_2O$ peak from the
smoke layer becomes discernible. Further up from the tropopause, when the peak $H_2O$
concentration is well above the extratropical tropopause at atmospheric pressure of less than 110
hPa, a deep minimum appears between the tropopause gradient and the peak from the smoke
layer. All $H_2O$ peaks were fitted with a Gaussian distribution operating on logarithmic pressure
and $H_2O$ concentration to obtain estimates of the peak concentration and the corresponding
atmospheric pressure. To investigate a time dependence in the smoke layer composition the peak
$H_2O$ concentration ($C_{H2O}$) was compared with the attenuation-corrected aerosol scattering ratio
(R) from CALIOP, the optical equivalent of the mixing ratio, where the latter was obtained by
forming the geometrical mean over 900 m around the peak scattering ratio. The ratio of the of the
two quantities ($R/C_{H2O}$) was formed, and its dependence on time from the fire was studied. Out of
the 13 smoke layers available with peak water vapor concentrations above the altitude of 110 hPa
atmospheric pressure, one was flagged as low quality in the MLS data set, leaving 12
observations for the study of the $R/C_{H2O}$ evolution.

332

### *2.5 UV aerosol index from OMPS-NM*

334

The UV aerosol index of OMPS-NM based on measurements at two wavelengths, 340 and 378.5
nm, is the official NASA aerosol index product according to OMPS-NM (NMMIEAI-L2 V2.1.1)
release notes (Torres, 2019). For strongly UV absorbing aerosols, like black carbon from
wildfires, the UV aerosol indexes strongly increases with altitude (Herman et al., 1997). Here the
OMPS-NM UV aerosol index was used to map the geographical evolution of the smoke layers,
that according to CALIOP measurements were distributed in both the troposphere and the
stratosphere.

342

## 3. Results

344

Here we use an approach based on five satellite sensors to study the influence on the stratosphere
of the great North American fire in August 2017. We start by briefly describing results from the
method to correct CALIOP data for attenuation of the backscattered laser light. Then follows a
comparison of AODs obtained from OMPS-LP and CALIOP. Absorption aerosol index from



OMPS-NM is used to describe the dispersion of the wildfire aerosol in the stratosphere. To
explain differences in AOD between OMPS-LP and CALIOP, a comparison of extinction
coefficients follows, where results from SAGE III/ISS also are included in the comparison. The
evolution of the optical properties of the wildfire aerosol is then described, before the North
American wildfire aerosol is compared with volcanic influence on the stratospheric AOD.
Finally, the fifth data set, water vapor from the MLS, is introduced in the discussion section,
where the evolution of the wildfire aerosol in the stratosphere is analyzed.

### 3.1 Correction for attenuation

The smoke layers usually were 1 – 3 km thick and could extend several degrees in longitude and
latitude. Measurements with the CALIOP lidar provide, in addition to short, nadir-viewing
measurement path in dense layers, the advantage that the signal is retrieved as a function of
position along the laser path with high resolution, which can be used to correct for attenuation of
the signal. Figure 3a shows the attenuated scattering ratio (R'; the measured backscattering
divided by the calculated molecular backscattering) from an example-smoke-layer measured on
August 16, 2017. The scattering ratio should be close to 1 in air layers with low aerosol
concentration, whereas values below 1 is caused by attenuation from particles. As can be seen in
Figure 3a, the attenuated scattering ratio first increases (starting from above the layer). Then the
signal decreases and reaches well below unity from 11 km altitude and downwards, i.e., well
below the scattering ratio of particle-free air. By techniques described in the Methods section we
correct for attenuation and fit the lidar ratio (Figure 2a) to obtain an estimate of the
backscattering without attenuation, as illustrated by the scattering ratio (R) in Figure 3a.

The evolution of wildfire aerosol from day 3 to 59 after the North American PyroCbs on August
12, 2017, is first investigated by comparing 32 smoke layers from individual CALIOP swaths.
The influence from attenuation is shown in Figure 3b. Clear deviation from the 1:1 line appears
already at layer attenuated (uncorrected) AODs ($AOD_{att}$) of 0.12, and 50% reduction of the signal
appears at layer $AOD_{att}$ of approximately 0.25. Reduction by more than 50% appears until day 10
after the fire, whereas those measurements close to the 1:1 line were taken after day 30. The
AOD, i.e., the AOD corrected for attenuation, exceeds the $AOD_{att}$ by more than a factor of 5 in
the densest layers of this study (Figure 3b).

### 3.2 Comparison of CALIOP and OMPS-LP

To study the evolution of the stratospheric AOD, we form a 3-dimensional box in the
stratosphere extending over all longitudes in the 20 – 80° N latitude range. In this box we use all
daily profiles ,14 – 15 CALIOP and 42 – 45 OMPS-LP, to form the average AOD. We apply the
method to correct CALIOP data for attenuation, as described in the Methods section. AODs are
computed for three layers, the LMS, the lower Brewer-Dobson branch, and the upper Brewer-
Dobson branch, as shown in Figure 5.





When comparing AODs, the measurement wavelengths should be as close as possible, due to the
wavelength dependence of scattering. CALIOP AODs are shown for 532 nm wavelength, and the
OMPS-LP data are shown for the close wavelength of 510 nm. In addition, the 745 nm AODs
from OMPS-LP is shown. The response to the 2017 North American fire is weak in the upper
Brewer-Dobson branch (Figure 5a), whereas the two lower layers (Figures 5b-c) show clear
increase of the AOD. Comparing the two methods, they agree well in the upper Brewer-Dobson
branch. In the lower Brewer-Dobson branch we see good agreement between the two methods,
except for the first 1 – 2 months after the fire where much higher AODs are recorded by CALIOP
(Figure 5b). The latter is also true for the LMS, whereas the general agreement between the two
methods is poor (Figure 5c). The OMPS-LP documentation advise against using data from below
approximately 17 km altitude, approximately the upper limit of the LMS, due to loss of
sensitivity (Taha et al., 2020). We therefore do not perform any further comparisons in the LMS.
The stratosphere above the LMS (above the 380 K isentrope) shows good agreement between the
two methods, except for the first 1 – 2 months after the fire (Figure 5d).

*3.3 Early evolution of the smoke layers*

The daily AOD averages show large variability the first days after the fire because the lidar
measures narrow curtains through the atmosphere, Figure 5e. The variability remains until the
smoke layers become sufficiently dispersed, allowing several daily measurements of the smoke
layers. The nadir-viewing OMPS-NM provides UV (ultraviolet) absorbing aerosol index, where
strong signal for strongly UV light absorbing aerosol is obtained in the upper troposphere and the
stratosphere. Figure 6 shows the geographic evolution of the smoke layers from August 14 to 22,
2017 together with the orbits followed by the CALIOP measurements. Up to August 16 the
smoke is found in a rather confined area. From August 17 the smoke layers are stretched in
Easterly direction, and after that the smoke spreads rapidly to the East. The dispersion gradually
increases the number of daily CALIOP observations of the smoke. This can also be seen in
Figure 5e, where the variability in the daily AOD data becomes successively smaller. From day
10 (August 22) we see a clear pattern of decline of the AOD.

Figure 5e shows the total stratospheric AOD according to CALIOP from the tropopause to 35 km
altitude. We see a strong decline of the stratospheric AOD the first 1.5 months after the fire, and a
fitted exponential function has a half-life of $6.5 \pm 0.9$ days. Such a decline cannot be found in the
OMPS-LP AODs, which instead are increasing during the first month.

To further investigate this clear difference between the two methods, individual smoke layers are
investigated with respect to extinction coefficients. Figure 7a-d show the extinction coefficient of
strong smoke layers from four days in August and September 2017. From CALIOP we show the
attenuated extinction coefficients as well as the profiles corrected for attenuation. Together with
the CALIOP data the OMPS-LP data closest by are shown. It is obvious that OMPS-LP shows
very much smaller reaction to the smoke layers than CALIOP. However, we cannot be sure that
the two instruments viewed the same airmasses in these four examples, because the two
instruments do not belong to the same satellite constellation. To remove that obstacle, the daily



maximum stratospheric extinction coefficient from OMPS-LP was extracted and compared with
32 selected profiles' peak extinction coefficients from CALIOP. SAGE III/ISS was also included
in the comparison from day 19 after the fire. Unfortunately, the orbiting of ISS did not permit
measurements of the fire studied here before that day. The very strong signals from CALIOP are
not reflected in the OMPS-LP or SAGE III/ISS measurements, see Figure 7e. In part, this can be
explained by difference in vertical resolution, but as shown in Figures 7a-d, these high extinction
coefficients extend to broad vertical ranges that should allow detection of strong signals also by
OMPS-LP and SAGE III/ISS.

There is one principal difference between CALIOP on one hand and OMPS-LP and SAGE
III/ISS on the other hand: whereas the former is nadir-viewing (vertical) the latter two methods
operate in limb orientation (horizontal). This is important, because the horizontal extension of
smoke layers is much larger, e.g., the smoke layer in Figure 1b has a vertical extension of
approximately 2 km, whereas the horizontal extension is approximately 700 km. The vertical,
two-way transmission to the CALIOP sensor through this layer is approximately 0.01, which we
correct for. The horizontal path through this layer is 350 times longer, implying that the one-way
limb transmission becomes $10^{-350}$ for the same wavelength. Even if the horizontal extension
would be just one tenth the transmission is still as low as $10^{-35}$. Obviously, the radiation used for
detection in OMPS-LP and SAGE III/ISS is rapidly eliminated in such smoke layers. Therefore,
these two methods are inadequate for studies of dense aerosol layers. The upper limit in terms of
vertical AOD is estimated to 0.02 (SAGE III/ISS Users Guide, 2018), corresponding to the
extinction coefficient of 0.02 km$^{-1}$ for a 1 km thick layer. This problem is also acknowledged for
OMPS-LP (Chen et al., 2018; DeLand, 2019). Despite the clear limitation of OMPS-LP and
SAGE III/ISS in this respect, the large body of information on wildfires is based on these
methods, e.g., Bourassa et al., (2019), Das et al., (2021), Khaykin et al., (2020), Kloss et al.,
(2019), Torres et al., (2020) and Yu et al., (2019). By comparing with CALIOP we here show
that the limb-oriented techniques miss the dramatic events during the first 1 – 2 months after the
fire. The rapid decline of the wildfire smoke will be further analyzed below.

### *3.4 Aerosol optical properties*

To further investigate the unusual evolution of the AOD, we turn to the optical properties of the
wildfire aerosol. The particle color and depolarization ratios are shown in Figure 2b and c. To test
the significance in the evolution the data were temporally divided into two equal halves by
number of data points, and geometric averages were formed (black lines in Figure 2). The particle
color ratio shows a highly significant decrease comparing the first to the last half of the data
points, whereas the particle depolarization ratio increases with high significance. The change in
the optical properties takes place up to 15 – 30 days after the fire. This coincides with the decline
of the AOD, thus connecting a change of the aerosol properties to the AOD decline.

### *3.5 Stratospheric AOD variability caused by volcanism and wildfires*





The stratospheric AOD varies considerably over time mainly due to influence from explosive
volcanic eruptions as demonstrated in Figure 8, showing the period 2008 – 2018. In this time
span, nine volcanic eruptions clearly, but to varying degree, affected the stratospheric AOD. We
also identify two cases of influence from wildfires, the Victoria fire (Australia, 2009) and the fire
studied here (Western North America, 2017). The residence time in the stratosphere varies from
several years for tropical injections into the upper layer representing the upper branch in the
Brewer-Dobson circulation (BD) (Figure 8a), the order of a year in the shallow branch of the BD
circulation (Figure 8b), to months in the LMS (Figure 8c) (Friberg et al., 2018). The sum of the
three layers is shown in Figure 8d. The volcanic eruptions in these 11 years mainly affected the
two lower stratospheric layers, only the Kelut eruption (2014) clearly reached to the deep BD
branch. Fire aerosol contains black carbon, which absorbs radiation, heats surrounding air and
induces lifting, as observed after the fire studied here (Khaykin et al., 2018; Yu et al., 2019).
After both fires, we see weak AOD elevation in the deep BD branch (Figure 8a), but for the fire
studied here the two lower layers dominate the AOD, like most of the volcanic eruptions in the
eleven-year period.
Comparing the evolution of the AOD of the North American wildfire with the evolution of the
aerosol from two of the most important volcanic eruptions during the last 25 years (Figure 9), we
find that the maximum stratospheric AOD after the fire is similar to that after the 2011 Nabro and
2009 Sarychev eruptions. During the first couple of months after volcanic events the AOD grows
due to formation of condensable sulfuric acid from the emitted volcanic gas sulfur dioxide. In
contrast, the wildfire aerosol displays a rapid decline during the first few weeks, before the AOD
stabilizes (Figure 9). This is followed by a period of rather stable AOD of more than 6 months,
before the AOD evolution turns to a slower decline towards background conditions, with similar
seasonality as the aerosol from the volcanic eruptions discussed (Figure 9). This latter decline is
mainly caused by springtime transport out from the stratosphere at mid and high latitudes
(Bönisch et al., 2009; Martinsson et al., 2017).
**4. Discussion**
The smoke aerosol is distributed both in the LMS and in the lower BD branch like aerosol from
several volcanic eruptions (Figure 8). The rapid decline of the smoke aerosol during the first
month after the fire thus cannot be explained by transport out of the stratosphere. Measurements
with Raman lidars at three wavelengths indicate that the smoke from this North American fire
contain an accumulation mode but no coarse mode (Haarig et al., 2018; Hu et al., 2019). The
influence from sedimentation on submicron diameter particles is small (Martinsson et al., 2005).
Moreover, the change in the particle depolarization ratio (Figure 2c) indicates change of the
aerosol particle properties, and the particle color ratio decrease after the fire (Figure 2b) is the
expected outcome for reduced particle sizes. Based on these arguments we turn the attention to
loss of material from the aerosol particles to the gas phase to explain the rapid decrease in AOD
seen in Figure 5e.



Smoke layers contain water vapor that could induce hygroscopic growth/shrinkage. Water vapor
profiles for individual smoke layers from days 6 – 60 after the fire were obtained from the MLS.
Measurements close to the tropopause (Figure 10a) are affected by a steep gradient in $H_2O$
concentration. The profiles well above the gradient peaking at atmospheric pressure of less than
110 hPa are shown in Figure 10b. For the latter category the peak $H_2O$ concentration is in the
range 7 – 14 ppmv, implying a maximum $H_2O$ vapor pressure of 0.16 Pa. For typical conditions
in the extratropics that vapor pressure corresponds to a relative humidity of a few percent or less
(Murphy and Koop, 2005).
To further investigate the smoke layers, the temporal evolution of the composition is studied by
forming the ratio of the mixing ratios of two components: aerosol backscattering and $H_2O$ at the
peak of respective vertical distribution. As pointed out above, the strong $H_2O$ gradient around the
tropopause affects the MLS measurements. But for the smoke layers higher up, peaking above
110 hPa, we find a rapid decrease in the aerosol scattering ratio compared with the $H_2O$
concentration (Figure 10c). Fitting an exponential function ($\frac{R}{C_{H_2O}} = a + be^{-t/\tau}$), the half-life
becomes 9.7±3.2 days, which is somewhat longer than that computed from the AOD (half-life
6.5±0.9 days). The rapid AOD decline (Figure 5e) is thus verified by relative concentrations of
aerosol and $H_2O$ under well-controlled humidity conditions, whereas the low relative humidity
rules out hygroscopic growth and influence from clouds as the explanation of the AOD decline.
The near-field wildfire aerosol contains, besides black carbon (Bond et al., 2013; Ditas et al.,
2018), approximately 90% organic material (Garofalo et al., 2019). After emission, secondary
organic aerosol (SOA) is formed by oxidation of gas phase compounds (Shrivastava et al., 2017).
Knowledge of processes controlling formation and removal in the atmosphere is limited (Hodzic
et al., 2016). Global aerosol models usually remove SOA mainly by wet (90%) and, to a smaller
extent, by dry deposition (Tsigaridis et al., 2014). In contrast to the species dominating the
stratospheric aerosol and its precursor compounds during background conditions and volcanic
influence (sulfuric acid and sulfur dioxide), organic species are not the ultimate
thermodynamically stable compounds (Hallquist et al., 2009). Organic aerosol is an intermediate
state on routes, with little known rates, from emitted compositions to the highly oxidized gaseous
products CO and $CO_2$ (Jimenez et al., 2009). Modeling and numerous laboratory studies find
evidence for photolytic removal rates of organic aerosol similar to that of wet deposition in the
troposphere (Hodzic et al., 2016; Zawadowics et al., 2020). Recently, photolytic removal of
particulate SOA was included in the Whole Atmosphere Community Climate Model (WACCM6)
(Gettelman et al., 2019). Hodzic et al. (2015) estimate the photolytic loss over a 10-day period to
50% for most organic species at mid tropospheric conditions.
These high rates are disputed by Yu et al. (2019), claiming a lifetime of 150 days (halflife 104
days) of organic aerosol from the fire studied here, whereas Das et al. (2021) explain a similar
half-life of the same fire by large-scale circulation and particle sedimentation using OMPS-LP
and modeling. The experimental data used here cannot differentiate these two explanations,
although the slow part of the smoke decline is similar in seasonality to that of volcanic aerosol
(Figure 9) where photochemical loss is less important. The modeling study by Yu et al. (2019)



was based on mimicking the extinction according to SAGE III/ISS at 1020 nm wavelength at 18
km altitude. For three reasons their study misses the strong decline of the AOD during the first
month. Firstly, because the orbiting of ISS prohibits studies of the wildfire smoke the first 19
days after the fire, secondly because of the time required to transport the wildfire aerosol to 18
km altitude is approximately one month (Yu et al., 2019) and thirdly because problems with
event termination ("saturation"), see Figure 7e. We therefore conclude that that Yu et al. (2019)
could not observe the main decline of the aerosol taking place during the first 1 – 2 months after
the fire, see section 3.3 for further details.
Submicron aerosol particles have much longer residence time in the stratosphere than in the
troposphere due to sparsity of clouds, thus inhibiting the sink that traditionally is considered the
most important in the troposphere, i.e., wet deposition. This provides unique possibilities to study
photolytic loss without competition from other aerosol sinks. Interpreting the body of evidence
on the strong and rapid decline of the stratospheric AOD during the first month after the fire, we
find that photolytic loss of organic aerosol is a highly likely explanation. The rate of photolytic
loss is likely better described by the evolution of $R/C_{H2O}$ than by the AOD, because the latter
could to some degree be affected by transport across the tropopause. Our strong experimental
evidence leads us to the hypothesis that the rapid decline of the wildfire aerosol in the
stratosphere with a half-life of 10 days is caused by photochemical loss of organic material. This
should be further investigated by modeling, but that is outside the scope of the present study.
To further put the strong early decline of wildfire aerosol into context, we compare the AOD
during background conditions (years 2013 and 2014) with the year of the fire. When the
contribution of the exponential term is very small of the wildfire aerosol (after 7 half-lives), the
background is approximately 2/3 of the wildfire AOD (Figure 9). Taking the background into
account, the excess stratospheric aerosol due to the wildfire declines by 83% from the $R/C_{H2O}$
value day 10 after the fire. The process starts before day 10, indicating that almost all the organic
aerosol constituting approximately 90% of the near-field wildfire aerosol mass (Garofalo et al.,
2019) could be lost by photolysis. Residual wildfire aerosol particles, likely stripped off by a
large fraction of its original organic content, remain in the stratosphere up to approximately one
year (Figure 9).
Finally, we investigate the stratospheric aerosol load from the wildfire by comparing with the
more studied volcanic impact (Table 1). The AOD growth, the average AOD over one year from
the fire/eruption subtracted by the average background AOD (2013 – 2014), is approximately 1/4
and 1/3 of that of two of the most important volcanic eruptions for the stratospheric aerosol in the
last 25 years (Sarychev 2009, Nabro 2011). The average excess aerosol during the year following
the fire corresponds to a radiative forcing of -0.06 W m$^{-2}$ in the region 20 - 80° N, using standard
conversion as an approximation (Solomon et al., 2011).
**Conclusions**



In this study we investigate massive injections of smoke into the stratosphere from the August
2017 North American wildfires using five satellite sensors. Methodology was developed to
correct CALIOP data for attenuation of the laser signal. The CALIOP AOD and extinction
coefficients were compared with OMPS-LP and SAGE III/ISS. From 1 – 2 months after the fire
we find that OMPS-LP and CALIOP AOD agree very well at altitudes above the 380 K
isentrope, where the former demonstrates high sensitivity with small statistical fluctuations. The
methods differ dramatically during the first 1 – 2 months after the fire when the smoke layers are
dense, because the long optical path through the smoke of the limb-oriented instruments OMPS-
LP and SAGE III/ISS cause event termination ("saturation"). This is clearly demonstrated by the
low daily maximum extinction coefficients of the two instruments, being orders of magnitude
lower than the peak extinction coefficients of CALIOP. The nadir viewing CALIOP experiences
a much shorter optical path, because the vertical extension of smoke layers usually are orders of
magnitude shorter than for limb orientation. We find that CALIOP is an indispensable tool for
studies of dense smoke layers entering the stratosphere after intense wildfires, providing signal
along the laser path that can be used to correct for attenuation. Once the smoke layers are
sufficiently thin, the limb technique OMPS-LP provide sensitive measurements of the AOD that
can be used together with CALIOP.
The AOD from the wildfire declines exponentially with a half-life of 6.5 days. This decline is
further studied by the evolution of the ratio of the aerosol and water vapor mixing ratios of the
smoke layers, resulting in a massive decline of 80 – 90% of the wildfire aerosol with a half-life of
approximately 10 days. We find transport out of the stratosphere, sedimentation, influence from
clouds or hygroscopic growth/shrinkage to be highly unlikely explanations for the rapid decline
of wildfire aerosol in the stratosphere. Based on strong experimental evidence we hypothesize
that photochemical loss of organic aerosol causes the rapid decline, which would mean that
almost the entire organic fraction of the wildfire aerosol would be lost in the exponential decline.
The half-life according to this study agrees well with results from laboratory studies and global
modeling. Our unique result could be obtained because of the long residence time of aerosol
particles in the stratosphere, whereas tropospheric studies of photochemical loss are extremely
difficult because it is masked by SOA formation and wet deposition due to short residence time.
The residual aerosol leaves the stratosphere within a year in the Brewer-Dobson circulation.
Despite the initial loss, the long-term effects of wildfire smoke on the stratospheric AOD and
radiative forcing are considerable. The ongoing climate change is projected to increase the
frequency of wildfires, prompting the need for inclusion of wildfire impact on the stratospheric
aerosol load in the climate models.
**Acknowledgements**
Aerosol products from the CALIOP sensor and SAGE III/ISS were produced by NASA Langley
Research Center. The official NASA aerosol index from the OMPS Nadir Mapper, the aerosol
scattering from OMPS Limb Profiler and water vapor profiles from MLS are supplied by
Goddard Earth Sciences Data and Information Services Center. We gratefully acknowledge
financial support from the Swedish Research Council for Environment, Agricultural Sciences and



Spatial Planning (contract 2018-00973), the Swedish National Space Board (contracts 130/15 and
104/17), and the Crafoord foundation (contract 20190690).

**Author Contributions**

B.G.M. designed the study, designed methodology, undertook part of the data analysis, and wrote
most of the paper. J.F. contributed to the design of the study, designed methodology, did part of
the data analysis, and wrote parts of the text. O.S.S. contributed to the data analysis and M.K.S.
contributed to the design of methodology. In addition, all authors participated in discussions and
commented on the manuscript.

**Data availability**

CALIOP V4.10 lidar data (https://search.earthdata.nasa.gov/search?fp=CALIPSO) are publicly
available.
OMPS-NM UV aerosol index was obtained from the publicly available site
https://worldview.earthdata.nasa.gov/.
OMPS-LP stratospheric aerosol optical depths were obtained from
https://disc.gsfc.nasa.gov/datasets/OMPS_NPP_LP_L2_AER_DAILY_2/summary
MLS water vapor concentrations were obtained from
https://disc.gsfc.nasa.gov/datasets?page=1&keywords=ML2H2O_005
SAGE III/ISS aerosol data were obtained from
https://asdc.larc.nasa.gov/project/SAGE%20III-ISS/g3bssp_51.

**Competing Interest**

The authors declare no competing interests.

**Additional Information**

Correspondence and requests for materials should be addressed to B.G.M.

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






**Tables**

Table 1. Maximum and yearly average stratospheric AOD during background conditions and
during one year after the fire and after the two volcanic eruptions in Figure 9.

| Year | Background 2013 | Background 2014 | Wildfire 2017 | Sarychev 2009 | Nabro 2011 |
|---|---|---|---|---|---|
| AOD max | 0.009 | 0.009 | 0.020 | 0.028 | 0.017 |
| AOD | 0.0075 | 0.0074 | 0.0097 | 0.0169 | 0.0138 |
| AOD growth[a] | - | - | 0.0023 | 0.0095 | 0.0064 |
| RF[b] | - | - | -0.06 | -0.24 | -0.16 |

[a]Growth of AOD due to influence from wildfire/volcanism obtained by subtracting the average of 2013
and 2014 AOD.
[b]Radiative forcing (W m$^{-2}$) of the background-subtracted AOD.



**Figures**

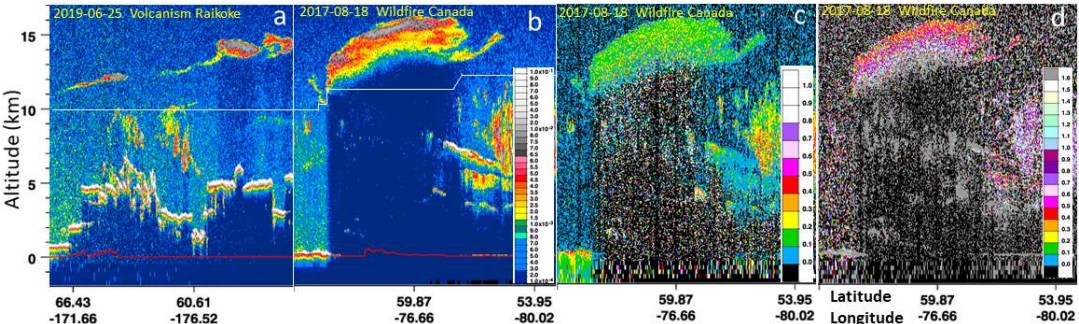



Figure 1. CALIOP curtains of total attenuated backscatter (km$^{-1}$ sr$^{-1}$) at 532 nm from a) volcanic
aerosol layers in the stratosphere three days after the 2019 Raikoke eruption and b) a
stratospheric smoke layer from the August 12, 2017, North American wildfire. c) Volume
depolarization ratio at 532 nm and d) attenuated color ratio (1064 to 532 nm) for the curtain in b).
The white lines in a) and b) show the position of the tropopause.

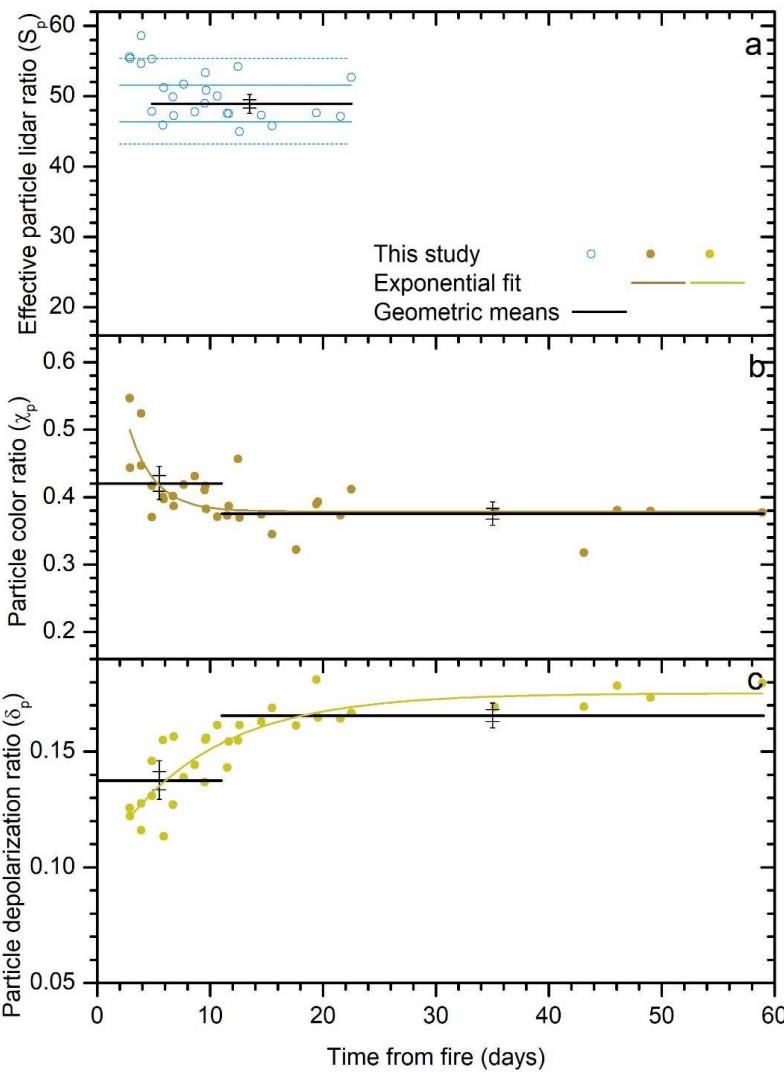

Figure 2. Particle optical properties during the first 60 days after the fire. Black error bars show standard error and the double-sided 95% probability range of the geometric means. a) Particle lidar ratios for 532 nm where data points with fitting error exceeding 25% are discarded. The black line shows the geometric mean after day 4, and the full and dotted blue lines show the standard deviation and the double-sided 95% probability range of the distribution. b) Particle color ratio (1064 nm divided by 532 nm wavelength backscattering) with exponential fit ($R^2$ = 0.48, P < $10^{-10}$), and c) particle depolarization ratio with exponential fit ($R^2$ = 0.76, P < $10^{-10}$). The color and depolarization ratios were divided in two equal groups by number of observations to illustrate the highly significant changes with time of the optical properties, where the long and short error bars are the standard error and the double-sided 95% probability range of the geometric means.




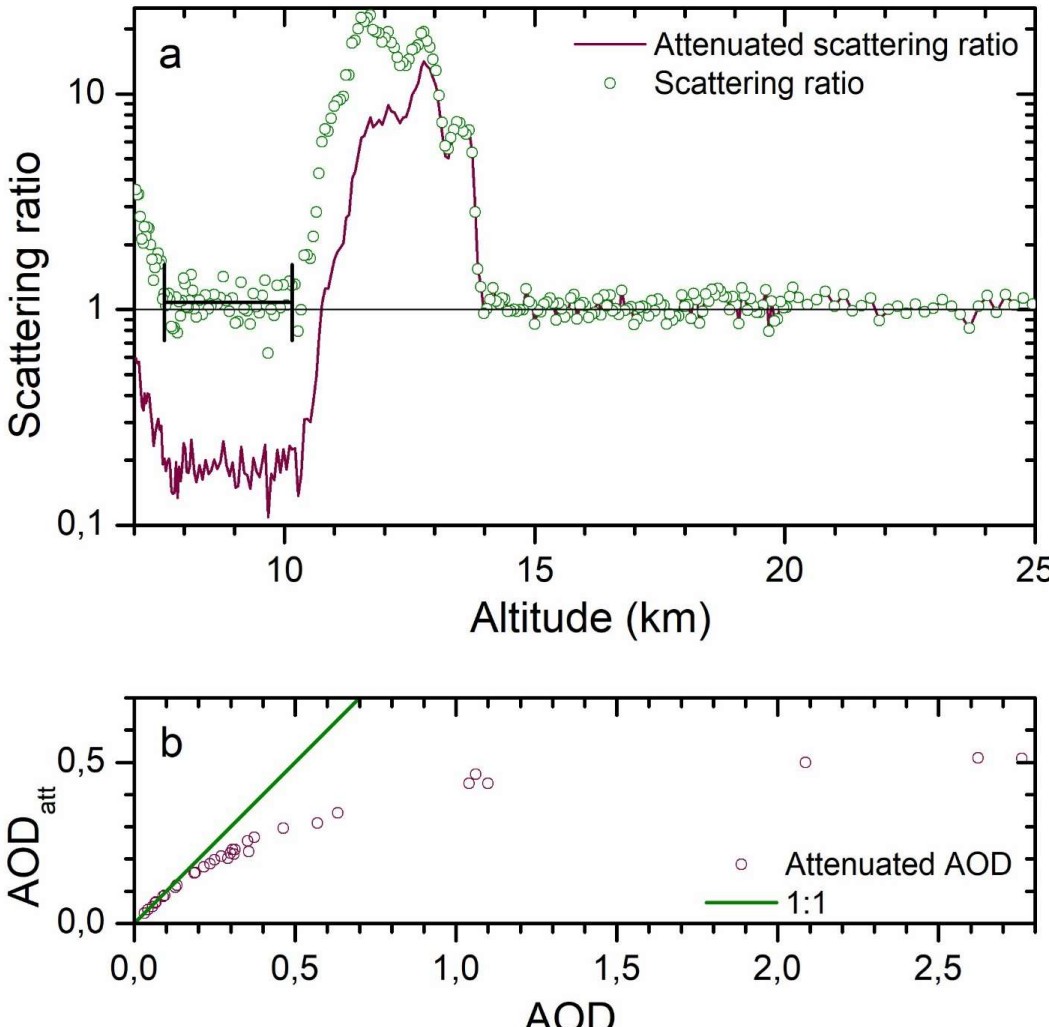

Figure 3. Illustration of methodology and its effect. a) The attenuated and corrected scattering
ratios as a function of altitude. Example of methodology for one smoke layer, where the
scattering ratio between 7.5 – 10 km altitude, below the smoke layer at 10.5 – 14 km, is targeted
to a value of 1.08 (explained in the method section) by iteratively fitting the lidar ratio for 532 nm
wavelength. b) The attenuated layer AOD ($AOD_{att}$) related to the layer AOD corrected for
attenuation. The 1:1 relation is shown by the full line.





Figure 4. OMPS-LP layer AODs averaged over 20 to 80° North for 745 nm wavelength using
data filtered and not filtered from clouds and polar stratospheric clouds, and with and without
data flagged for data quality. Layer AOD for a) the upper Brewer-Dobson branch (470 K
isentrope – 35 km), b) the lower brewer-Dobson branch (380 – 470 K) and c) the LMS
(tropopause – 380 K) are shown.

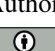

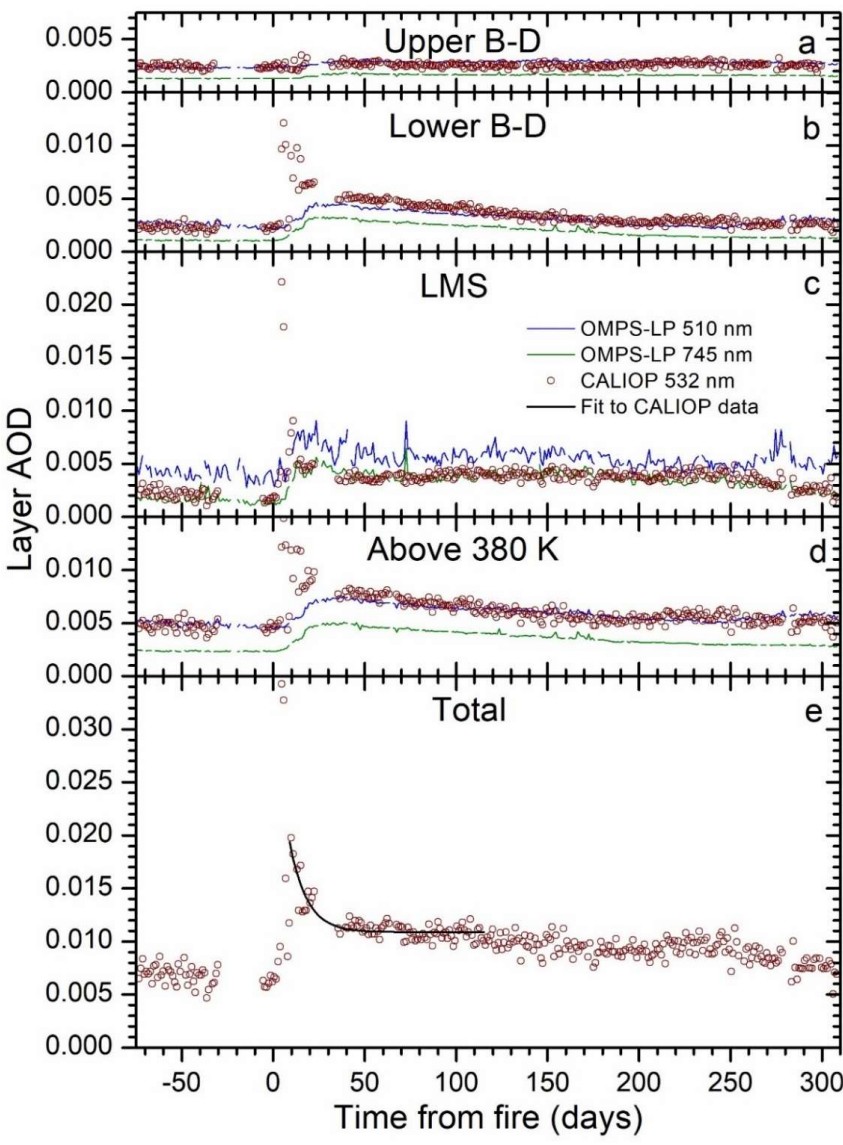


Figure 5. AOD evolution of the stratospheric AOD (daily average) from 75 days before to 310
days after the 2017 western North American fires. Comparisons of AOD from CALIOP (532 nm)
with OMPS-LP (510 and 745 nm) with cloud filtering and flags activated for a) the upper
Brewer-Dobson branch (470 K isentrope – 35 km, b) the lower Brewer-Dobson branch (380 –
470 K) c) the LMS (tropopause – 380 K) d) from 380 K to 35 km (sum of layers in a and b) and
e) the stratosphere of CALIOP from the tropopause to 35 km (sum of layers in a, b and c). The
black, full line is an exponential fit ($R^2 = 0.79$, $P < 10^{-10}$) to the AOD over days 10 – 115 after the
fire. The total stratospheric AOD half-life of the fit is 6.5±0.9 days.



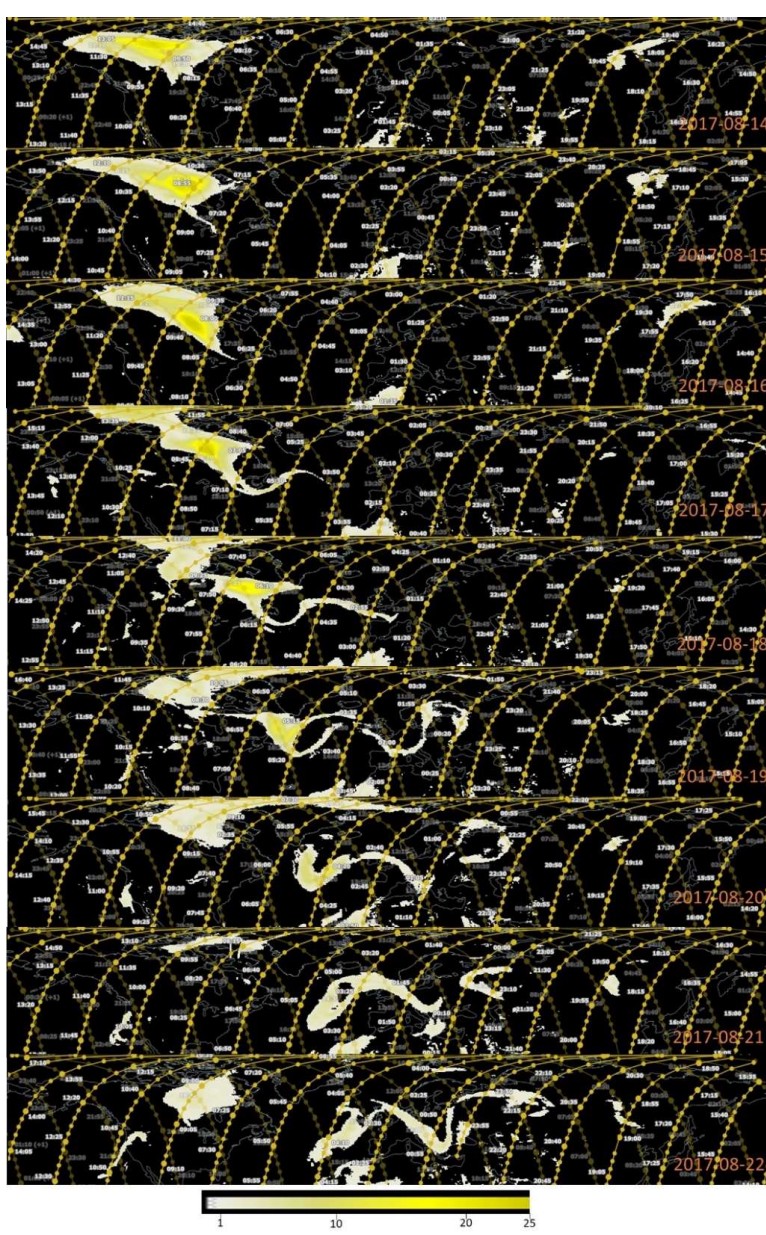

1015

Figure 6. Daily OMPS-NM aerosol absorbing index (UV) August 14 – 22, 2017 over all
longitudes and latitudes 20 - 80° N. This index is sensitive to UV absorbing aerosol particles in
the upper troposphere and the stratosphere, where signals from tropospheric aerosol declines
faster than from stratospheric due to short residence time. The yellow lines indicate nighttime
swaths of the CALIPSO satellite, and the faint lines show CALIPSO daytime swaths.




Figure 7. Extinction coefficients according to CALIOP, OMPS-LP and SAGE III/ISS in the 20 -
80° North latitude range during the first 60 days following the North American fire. a – d)
selected profiles (attenuated and corrected for attenuation) from CALIOP compared with closest
profiles according to OMPS-LP. e) Peak extinction coefficient from selected CALIOP profiles
compared with daily maximum extinction coefficients from OMPS-LP and SAGE III/ISS. Note
that SAGE III/ISS data are missing the first 19 days because of irregular coverage of the latitude
range of interest.



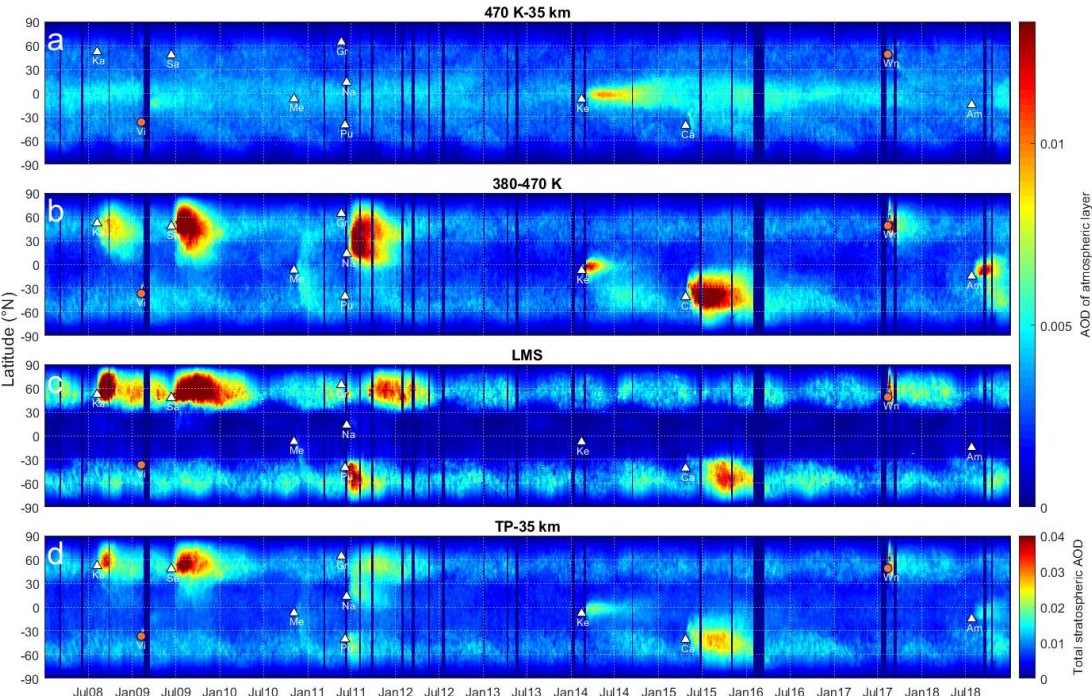



Figure 8. Zonally and eight-day moving average aerosol optical depth (AOD) of the stratosphere.
a - c) AOD in three layers obtained from CALIOP data (level 1B): a) 470 K potential temperature
to 35 km (deep Brewer-Dobson branch), b) 380 – 470 K (shallow Brewer-Dobson branch), c) the
tropopause to 380 K (LMS). d) The total AOD from the tropopause to 35 km altitude. Volcanic
eruptions marked by white triangles: Kasatochi (Ka), Sarychev (Sa), Merapi (Me), Grimsvötn
(Gr), Puyehue-Cordón Caulle (Pu), Nabro (Na), Kelut (Ke), Calbuco (Ca), and Ambae (Am), and
wildfires marked by orange circles: Victoria fire (Vi) and Western North American fires (Wn) at
time and latitude of eruption/fire. The AODs are corrected for attenuation.





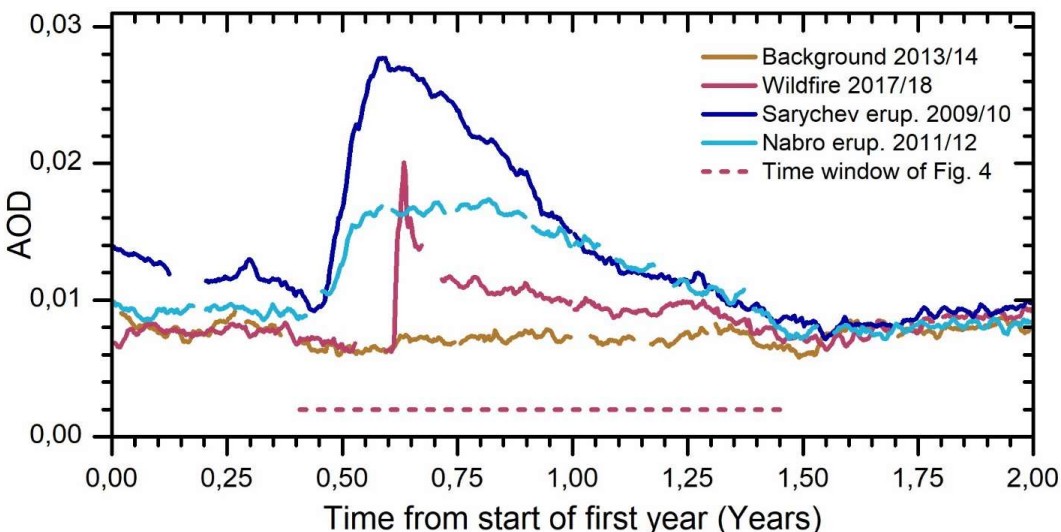


Figure 9. Evolution of the AOD in the 20 - 80° N interval (8-day moving average) over two
years: close to background conditions in the latitude interval studied (2013 – 2014), the year and
the following year of the August 12, 2017, fire (2017 – 2018), and the same for two volcanic
eruptions, the June 12, 2009, Sarychev (2009 – 2010) and June 12, 2011, Nabro (2011 – 2012)
eruptions.

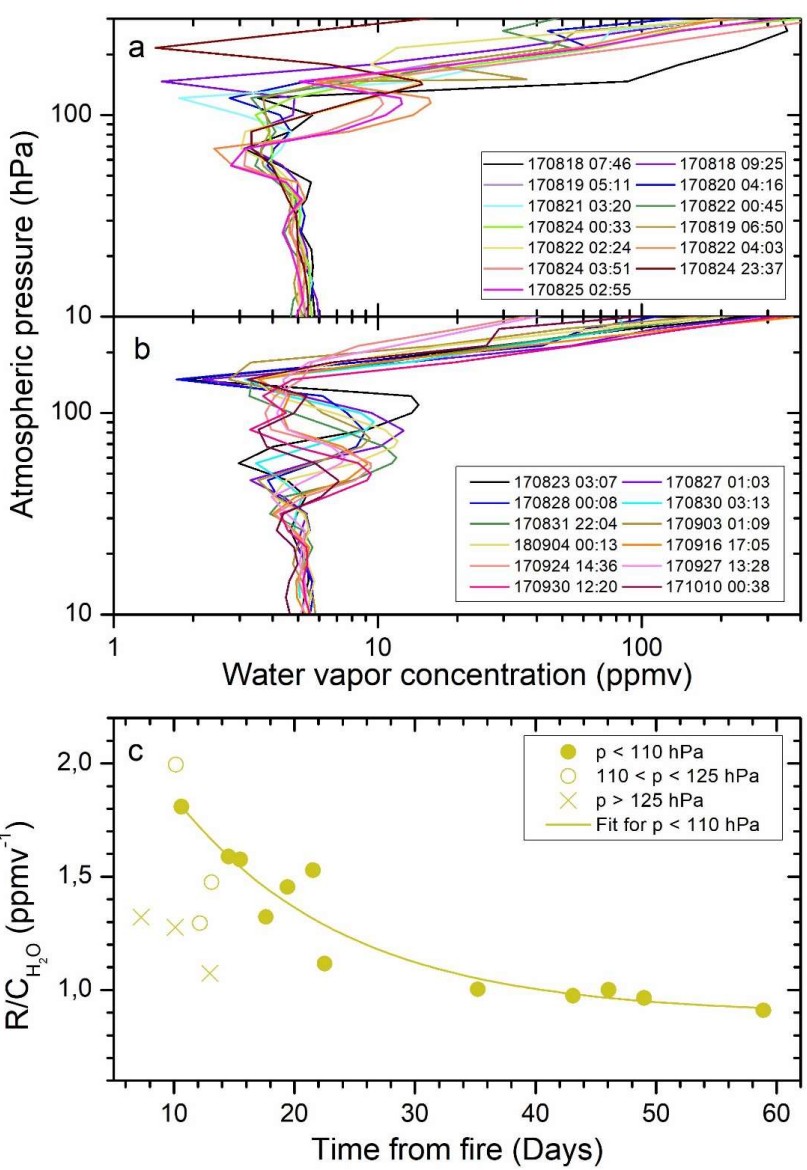

Figure 10. Water vapor in the smoke layer. Microwave Limb Sounder (MLS) measurements of water vapor concentrations (ppmv) Vs. atmospheric pressure for smoke layers a) close to the tropopause and b) well above the tropopause (atmospheric pressure < 110 hPa at the $H_2O$ peak) for individual smoke layers available days 6 – 60 after the fire. c) The peak scattering ratio (R) according to CALIOP divided by the peak water vapor concentration ($C_{H2O}$) from MLS. The full line is an exponential fit ($R^2 = 0.88$, $P < 3 \times 10^{-10}$) to smoke layers peaking in water vapor concentration at a pressure less than 110 hPa. The half-life of the fit is 9.7±3.2 days.