# Peer review of "Five satellite sensor study of the rapid decline of wildfire smoke in the stratosphere"

_Atmospheric Chemistry and Physics, 2021_

## Referee Comment (RC1)

**Review of "Five satellite sensor study of the rapid decline of wildfire smoke in the stratosphere" by Martinsson et al.**

January 5, 2022

**1 General Comments**

This paper studies the aerosols injected into the stratosphere by the pyroCb event of August 2017 (named the "Pacific Northwest Event" (PNE) by several authors). I note that I am very familiar with this event, being the lead author on a recent paper on the subject. However, I can not claim great expertise in the measurement of aerosols.

The authors have done something which appears novel to me, in that they have combined both passive limb and active (Lidar) nadir data to show that:

- The two observation techniques do not agree when the aerosol is very thick because the limb path becomes opaque, and much of the aerosol in it is not observed.

- The aerosol decays very rapidly during the first week or two, staying relatively constant after that. The authors hypothesise that the aerosol contains two components: black carbon, which is long-lasting, and organic particles, which are removed by photolysis on a short timescale.

The paper is generally well-written and is not too hard to follow. I note a few corrections to the English in the "Technical corrections" section below, but the errors I correct do not cause the writing to be hard to understand.

The figures could do with a considerable amount of improvement. I note some specific issues below, but I also note that the size of text used on the figures needs consideration throughout. I cannot always be sure what needs to be done as it is not always clear whether the figures are intended to be printed at single-column or at two-column width. The authors should aim to use text on their figures which will be of a similar size to the caption text.

It is a bit self-serving for me to suggest that the authors should reference my own paper on the 2017 event (`https://doi.org/10.5194/acp-21-16645-2021`), which also makes use of the MLS $H_2O$ data. However, a big lesson which I learned in the review process of that paper is that although August 2017 is over four years ago now, papers on the event continue to appear in the literature, and the authors will probably need to add a number of references, both while replying to the referees and while working with the production staff on the final copy. Papers they may feel the need to add include

- Lestrelin et al., (2021) (`https://doi.org/10.5194/acp-21-7113-2021`)

- Fromm et al., (2021) (`https://doi.org/10.1029/2021JD034928`)

The authors note that the event observed here is one of two very large events in the last few years, the other being the Australian New Year event of December 2019. The "Black Saturday" event of February 2009 was also quite large and also occurs within the operational period of CALIOP and MLS. I am not going to suggest that the authors should extend their analysis to either, or both, of these events. But they might spell out why they have not done so, and whether they intend to do so in the future.

**2   Specific Comments**

- Lines 12–13: To describe the source of the aerosols in this event as being in "Western North America" is a bit vague. A variety of studies (including my own, noted above) have pinned down the source region with more accuracy than this. (Also, "Western" here is not part of a name, so it should not have a capital letter.)

- Line 313: On mentioning the A-train it is probably a good idea to include a reference explaining what the A-train is. One possible reference is Tristan S. L'Ecuyer and Jonathan H. Jiang "Touring the

atmosphere aboard the A-Train", Physics Today **63(7)**, 36 (2010), doi:10.1063/1.3463626

- Line 970, Figure 1: The labelling on the colour scales in this figure is FAR too small, even if the figure is printed at two-column width.

- Line 1015, Figure 6: The tiny numbers on the CALIPSO orbit tracks are too small to read and not useful to the reader. They should be removed.

- Line 1029, Figure 8: This is quite a useful summary figure, but the authors should consider an alternative colour scale. This, as far as I can tell, is the notorious "jet" colour scale. See `https://doi.org/10.1038/s41467-020-19160-7` for a recent discussion of colour scales. If the authors are dead set on a scale with similar colours to jet they might try Google's "turbo" scale: see `https://ai.googleblog.com/2019/08/turbo-improved-rainbow-colormap-for.html`. It is not clear to me that the labels used for the volcanoes and the two fire events will be readable in the final figure; the authors should consider making these labels larger, and perhaps using a heavier font.

- line 1045, Figure 10: It is confusing that parts a and b are plotted with higher altitude at the bottom. The caption should explain that the legend items in a and b are dates in the form yymmdd. It might be worth reducing clutter by removing all of the year digits as they are 17 in every case.

**3  Technical Corrections**

- L56: "example on" should be "example of".

- L164: "The first weeks" should be "During the first weeks".

- L228, eq 5: It is better to avoid whole words (such as "base" and "top" as used here) in equations. If they are to be used, they should not be in italics.

- L262: "laps" should be "orbits". I would also suggest that "14 – 15 orbits" is misleading, suggesting that the satellite's speed is variable.

In reality, it completes exactly the same number of orbits per day, every day, but this number is not an integer, lying somewhere between 14 and 15. (This last point also applies to line 116.)

- L289: "likely" should be "probably". ("Likely" is an adjective synonymous with "probable", **not** an adverb synonymous with "probably".)

- L315: "Livesley" should be "Livesey".

- Line 408 "...variability the first ..." should be "...variability during the first ..."

- "Easterly" should be "an easterly". (Note that compass directions only have a capital letter when they are part of a name, such as North Dakota or East Timor. "East" later in the same sentence should also not have a capital E.) It might actually be better to use "eastward" rather than "easterly" due to the way that meteorologists use "easterly" to mean "coming from the east".

- line 993, Figure 3, line 1039, Figure 9, line 1045, Figure 10 and possibly elsewhere: As ACP is a journal in English, decimal points in axis labelling should be points, NOT commas.

- Line 1000, Figure 4. The notation 1E-3 should be avoided in the axis labels if possible. Either 0.001 or $10^{-3}$ would be an improvement.

- lines 658–667: ACP prefers that data sets are referenced with a DOI and an item in the references list. For example, the MLS water vapour data has doi:10.5067/Aura/MLS/DATA2508 and approved reference text "Lambert, A., Read, W. and Livesey, N. (2020), MLS/Aura Level 2 Water Vapor (H2O) Mixing Ratio V005, Greenbelt, MD, USA, Goddard Earth Sciences Data and Information Services Center (GES DISC), Accessed: [Data Access Date]".

- lines 679 – 957: The references should be consistent about how the DOI is presented. For preference it should always appear as doi:10.1029/2010GL042815 and never as https://doi.org/10.1029/2010GL042815 ; the authors currently have a mix of these two styles.

---

## Referee Comment (RC2)

**Review of manuscript *Five satellite sensor study of the rapid decline of wildfire smoke in the stratosphere* (acp-2021-1015) by Martinsson et al.**

**Summary**

This manuscript utilizes data from five satellite sensors to document the lifetime of the August 2017 Pacific Norwest pyroCb event and suggest photolysis as the mechanism responsible for the rapid aerosol mass decrease observed by satellite observations. Aerosol extinction measurements by CALIPSO-CALIOP, S-NPP OMPS Limb Profiler, and ISS-SAGE III are used in the analysis.

The paper is suitable for publication in ACP after the authors address the minor issues discussed below.

**General Comments**

Most of the manuscript is dedicated to the analysis of CALIOP's level1b data. No specific reason is given as to why the authors decided not to use the standard CALIOP level2 products but to develop their own interpretation of CALIOP measured backscattered radiances.

It should be emphasized that CALIOP's main advantage over SAGE III and LP observations is the availability of nighttime observations that, unlike SAGE III and LP, allowed for aerosol measurements during polar night. The usefulness of CALIOP's nighttime observational capability clearly comes across in this work where 29 out of 32 analyzed CALIOP profiles were nighttime observations.

The OMPS-LP - CALIOP AOD comparison yields no meaningful information on the accuracy of either measurement because of their implicit dependence on assumed aerosol properties.

Although photolytic destruction is a reasonable aerosol removal mechanism, the authors should address other possible mechanisms such gravitational settling. It could be argued that the initial massive injection included a variety of aerosol types and sizes some of which would have been removed by gravitational settling on time scales similar to that of photolysis.

**Specific Comments**

A key step in the retrieval of aerosol properties from lidar observations is the selection of the lidar ratio. The choice of lidar ratio involves specific assumptions on the polydispersion particle size distribution, particle shape and complex refractive index. Although, not explicitly stated in the manuscript, one can assume that the authors considered the implicit lidar ratio assumptions (and the associated aerosol model) in the standard CALIOP level 2 product (Omar et al., 2009) to be inadequate for the interpretation of CALIOP observations in the presence of stratospheric carbonaceous aerosols and, therefore, decided to carry out their own inversion of CALIOP's Level 1b data making use of an improved aerosol model. A brief description of both the standard and adopted aerosol properties should be presented along with the rationale leading to the re-interpretation of CALIOP level1b data.

The authors discuss the evolution of the lidar, color and depolarization ratios in detail. Although the technical definitions of these terms are well known to the lidar community, the manuscript fails to connect the variability of those parameters with the variability on the actual microphysical and optical properties of the aerosol layer of great interest to readers beyond the lidar the community. A discussion of results in terms of practically meaningful aerosol properties will enhance the science value of this contribution.

The manuscript includes several statements on assumed numerical values of parameters without any references. No doubt most lidar experts are familiar with those values, but references may be important for the at large aerosol community. Please provide references and/or a rationale for the quoted numerical values in the statements below

Line 135. ….When the depolarization ratio is less than *0.05* the data is considered background and the lidar ratio is set to *50* sr.

 Line 137. Ice-clouds were removed in the lowest 3 km of the stratosphere by identifying them in stratospheric layers where the backscattering was high (attenuated backscattering larger than *0.0025* km-1 sr-1).

Line 138-139 Data in these layers were classified as probable clouds if their δv was higher than *0.20*, or smoke if δv was between *0.05-0.20*..

Line 164 …..very dense with layer AODs exceeding 1.

Detailed observations of the rapid evolution of the stratospheric AOD during the first two weeks following the onset of the pyroCb were carried out by the DSCOVR-EPIC mapper and AERONET ground-based observations (Torres et al., 2020). Both measurements reported stratospheric AOD's significantly larger than 1.0.

Line 333. The UV aerosol index is available from a variety of UV-capable sensors on several platforms (Aura-OMI, SNPP-OMPS, DSCOVR-EPIC, S5P-TROPOMI). It should be pointed out that, when data on height of absorbing aerosol layers is available, the UVAI information content can be quantified in terms of the physically meaningful aerosol optical depth (AOD) and single scattering albedo (SSA) parameters. DSCOVR-EPIC near UV observations (Torres et al., 2020) were used to quantitatively (380 nm AOD and SSA) describe the 2017 Pacific Norwest pyroCb-triggered stratospheric aerosol layer on the first week following the injection using CALIPSO-provided aerosol layer height information.

Line 382. An objective evaluation of the accuracy of OMPS-LP -CALIOP measured aerosol extinction should be done using SAGE III as standard reference. Unlike CALIOP and OMPS LP aerosol extinction, SAGE III solar occultation observations require no aerosol model assumptions whatsoever. Although SAGE III measurements are spatially and temporally sparse, just a few collocations would be sufficient to assess the accuracy of the reported aerosol extinction products.

---

## Author Comment (AC1)

***Answer to the review of Hugh C. Pumphrey (reviewer 1).***

Thank you for the comments that have helped us improve the manuscript. You will find our answers in blue text below.

Review of "Five satellite sensor study of the rapid decline of wildfire smoke in the stratosphere" by Martinsson et al.

January 5, 2022

1 General Comments
This paper studies the aerosols injected into the stratosphere by the pyroCb event of August 2017 (named the "Pacific Northwest Event" (PNE) by several authors). I note that I am very familiar with this event, being the lead author on a recent paper on the subject. However, I can not claim great expertise in the measurement of aerosols.
The authors have done something which appears novel to me, in that they have combined both passive limb and active (Lidar) nadir data to show that:

• The two observation techniques do not agree when the aerosol is very thick because the limb path becomes opaque, and much of the aerosol in it is not observed.

• The aerosol decays very rapidly during the first week or two, staying relatively constant after that. The authors hypothesise that the aerosol contains two components: black carbon, which is long-lasting, and organic particles, which are removed by photolysis on a short timescale.

The paper is generally well-written and is not too hard to follow. I note a few corrections to the English in the "Technical corrections" section below, but the errors I correct do not cause the writing to be hard to understand.

The figures could do with a considerable amount of improvement. I note some specific issues below, but I also note that the size of text used on the figures needs consideration throughout. I cannot always be sure what needs to be done as it is not always clear whether the figures are intended to be printed at single-column or at two-column width. The authors should aim to use text on their figures which will be of a similar size to the caption text. It is a bit self-serving for me to suggest that the authors should reference my own paper on the 2017 event (https://doi.org/10.5194/acp-21-16645-2021), which also makes use of the MLS H2O data. However, a big lesson which I learned in the review process of that paper is that although August 2017 is over four years ago now, papers on the event continue to appear in the literature, and the authors will probably need to add a number of references, both while replying to the referees and while working with the production staff on the final copy. Papers they may feel the need to add include

• Lestrelin et al., (2021) (https://doi.org/10.5194/acp-21-7113-2021)
• Fromm et al., (2021) (https://doi.org/10.1029/2021JD034928)

The authors note that the event observed here is one of two very large events in the last few years, the other being the Australian New Year event of December 2019. The "Black Saturday" event of February 2009 was also quite large and also occurs within the operational period of CALIOP and MLS.

I am not going to suggest that the authors should extend their analysis to either, or both, of these events. But they might spell out why they have not done so, and whether they intend to do so in the future.

Thank you for the suggested references. We include them. We work on a manuscript on the 2019 – 2020 Australian fire which we aim to publish.

2 Specific Comments

• Lines 12–13: To describe the source of the aerosols in this event as being in "Western North America" is a bit vague. A variety of studies (including my own, noted above) have pinned down the source region with more accuracy than this. (Also, "Western" here is not part of a name, so it should not have a capital letter.)

We changed the spelling. The fires were around the border between Canada and USA, but mainly in Canada according to Fromm et al. (2021, JGR Atmospheres). We changed to a more detailed geographical description in the Introduction section but retain the shorter description in the abstract.

• Line 313: On mentioning the A-train it is probably a good idea to include a reference explaining what the A-train is. One possible reference is Tristan S. L'Ecuyer and Jonathan H. Jiang "Touring the atmosphere aboard the A-Train", Physics Today 63(7), 36 (2010), doi:10.1063/1.3463626

Thank you, we include the reference.

• Line 970, Figure 1: The labelling on the colour scales in this figure is FAR too small, even if the figure is printed at two-column width.

We agree, we have changed size. To save space we use the number format "aE-b" in one of the color scales. That way we will show as much as possible of the graph, and we explain the format in the figure caption.

• Line 1015, Figure 6: The tiny numbers on the CALIPSO orbit tracks are too small to read and not useful to the reader. They should be removed.

We agree in principle. However, this figure was generated from NASA Worldview where these numbers are not optional. We therefore want to keep the numbers, although we agree with the reviewer that the figure would look nicer without the numbers.

• Line 1029, Figure 8: This is quite a useful summary figure, but the authors should consider an alternative colour scale. This, as far as I can tell, is the notorious "jet" colour scale. See https://doi.org/10.1038/s41467-020-19160-7 for a recent discussion of colour scales. If the authors are dead set on a scale with similar colours to jet they might try Google's "turbo" scale: see https://ai.googleblog.com/2019/08/turbo-improved-rainbow-colormap-for.html. It is not clear to me that the labels used for the volcanoes and the two fire events will be readable in the final figure; the authors should consider making these labels larger, and perhaps using a heavier font.

We have not thought a lot about color scales before, but of course we became interested and tested a turbo scale. Particularly the distinct effects of turquoise and yellow in the jet scale we used produced steps that are not present in the turbo color scale. Therefore, we changed to a turbo scale in the revised version. Thank you for making us aware of this problem.

• line 1045, Figure 10: It is confusing that parts a and b are plotted with higher altitude at the bottom. The caption should explain that the legend items in a and b are dates in the form yymmdd. It might be worth reducing clutter by removing all of the year digits as they are 17 in every case.

We actually tried to reverse the scale, but the plotting software used somehow collapsed because the scale is logarithmic. Therefore, we ended up with the scale in this direction. Concerning the dates: we believe that the risk of misunderstanding the labels is less when including the year. We would therefore want to keep the labels as they are.

3 Technical Corrections

Thank you, we have made use of all your comments below.

• L56: "example on" should be "example of".
• L164: "The first weeks" should be "During the first weeks".
• L228, eq 5: It is better to avoid whole words (such as "base" and "top" as used here) in equations. If they are to be used, they should not be in italics.
• L262: "laps" should be "orbits". I would also suggest that "14 – 15 orbits" is misleading, suggesting that the satellite's speed is variable. In reality, it completes exactly the same number of orbits per day, every day, but this number is not an integer, lying somewhere between 14 and 15. (This last point also applies to line 116.)
• L289: "likely" should be "probably". ("Likely" is an adjective synonymous with "probable", not an adverb synonymous with "probably".)
• L315: "Livesley" should be "Livesey".
• Line 408 ". . . variability the first . . . " should be ". . . variability during the first . . . "
• "Easterly" should be "an easterly". (Note that compass directions only have a capital letter when they are part of a name, such as North Dakota or East Timor. "East" later in the same sentence should also not have a capital E.) It might actually be better to use "eastward" rather than "easterly" due to the way that meteorologists use "easterly" to mean "coming from the east".
• line 993, Figure 3, line 1039, Figure 9, line 1045, Figure 10 and possibly elsewhere: As ACP is a journal in English, decimal points in axis labelling should be points, NOT commas.
• Line 1000, Figure 4. The notation 1E-3 should be avoided in the axis labels if possible. Either 0.001 or 10−3 would be an improvement.
• lines 658–667: ACP prefers that data sets are referenced with a DOI and an item in the references list. For example, the MLS water vapour data has doi:10.5067/Aura/MLS/DATA2508 and approved reference text "Lambert, A., Read, W. and Livesey, N. (2020), MLS/Aura Level 2 Water Vapor (H2O) Mixing Ratio V005, Greenbelt, MD, USA, Goddard Earth Sciences Data and Information Services Center (GES DISC), Accessed: [Data Access Date]".
• lines 679 – 957: The references should be consistent about how the DOI is presented. For preference it should always appear as doi:10.1029/2010GL042815 and never as https://doi.org/10.1029/2010GL042815 ; the authors currently have a mix of these two styles.

---

## Author Comment (AC2)

*Answer to comment acp-2021-1015-CC1 from Albert Ansmann*

The method we developed to correct for attenuation of the lidar signal inherently corrects for multiple scattering. The transmission is estimated based on the effective lidar ratio, i.e., the lidar ratio is multiplied by the multiple scattering factor, which then is multiplied by the multiple scattering-affected backscattering (eqn 2 in the manuscript). We have clarified this in the revised manuscript. Thank you, hopefully we have now expressed this clearer in the manuscript.

OMPS-LP data are corrected for multiple scattering by model calculation, the total radiance error is estimated to 1-3% (Loughman et al., 2015). Thank you for this comment, we have added this information in the revised manuscript.

We are aware of the results by Haarig et al. (2018) and Hu et al. (2019). At one point we considered to estimate the multiple scattering factor and the true lidar ratio. Based on their results and literature consideration we found that the multiple scattering factor was approximately 0.85. However, the lidar ratio varies both between smoke layers and in time, and we did not measure in the same air masses as Haarig et al. or Hu et al. nor could we undertake comparisons over time. Therefore, we could not justify the estimate of the lidar ratio and multiple scattering factor. Instead, we stayed with the effective lidar ratio. Estimation of the multiple scattering factor and the true lidar ratio did not affect the AOD estimated by our method, see the first section above.

We did not assume a clean Rayleigh atmosphere ($R = 1$) below the smoke layers. Instead, we used the scattering ratio beside the smoke layer to obtain an estimate of aerosol load surrounding the smoke layer. On average the scattering ratio surrounding the different smoke layers investigated was $R = 1.08 \pm 0.05$. We used that scattering ratio as the target value in the fitting of the effective lidar ratio.

Thank you for your comments. We found them helpful in clarifying the meaning of our results.

Loughman, R., D. Flittner, E. Nyaku, and P. K. Bhartia, Gauss–Seidel limb scattering (GSLS) radiative transfer model development in support of the Ozone Mapping and Profiler Suite (OMPS) limb profiler mission. Atmos. Chem. Phys., 15, 3007–3020, doi:10.5194/acp-15-3007-2015, 2015.

---

## Author Comment (AC3)

Answers to reviewer 2

We thank reviewer 2 for sharing insights with the comments. Below follows our answers. Text of the reviewer in black and the authors in blue.

**Review of manuscript Five satellite sensor study of the rapid decline of wildfire smoke in the stratosphere (acp-2021-1015) by Martinsson et al.**

**Summary**

This manuscript utilizes data from five satellite sensors to document the lifetime of the August 2017 Pacific Norwest pyroCb event and suggest photolysis as the mechanism responsible for the rapid aerosol mass decrease observed by satellite observations. Aerosol extinction measurements by CALIPSO-CALIOP, S-NPP OMPS Limb Profiler, and ISS-SAGE III are used in the analysis.

The paper is suitable for publication in ACP after the authors address the minor issues discussed below.

**General Comments**

Most of the manuscript is dedicated to the analysis of CALIOP's level1b data. No specific reason is given as to why the authors decided not to use the standard CALIOP level2 products but to develop their own interpretation of CALIOP measured backscattered radiances.

1. We started to use CALIOP level 1B ten years ago. With support by a scientist from NASA we extended his studies of volcanism using CALIOP to also include the aerosol load in the LMS (Andersson et al., 2015; Nature Communications 6:7692, DOI: 10.1038/ncomms8692).

It should be emphasized that CALIOP's main advantage over SAGE III and LP observations is the availability of nighttime observations that, unlike SAGE III and LP, allowed for aerosol measurements during polar night. The usefulness of CALIOP's nighttime observational capability clearly comes across in this work where 29 out of 32 analyzed CALIOP profiles were nighttime observations.

2. We are not sure that we understand this comment. Compared with the limb-oriented methods we find the main advantages of CALIOP to be orders of magnitude shorter path through the smoke layers which is important to avoid event termination, and distinct signal along the laser path that can be utilized for the purpose of correcting the signal for attenuation. To that we can add extremely high vertical resolution.

The OMPS-LP - CALIOP AOD comparison yields no meaningful information on the accuracy of either measurement because of their implicit dependence on assumed aerosol properties.

3. In Fig. 2a we estimate the lidar ratio until day 22 after the fire, reaching the average 48.9 sr (95% confidence interval: 47.6 – 50.3 sr). Since that result does not deviate significantly from the typical stratospheric background (50 sr), we use 50 sr except for the densest layers (the first days after the fire) where any deviation in the fitted lidar ratio strongly affects the estimated scattering. After day 22 the smoke layers are too thin for the method to estimate the lidar ratio. We agree that the OMPS-LP results are based on standardized assumptions on the optical properties of the aerosol. The good agreement of OMPS-LP with CALIOP stems from that the estimated lidar ratio happens to be close to that of the typical stratospheric background aerosol.

Although photolytic destruction is a reasonable aerosol removal mechanism, the authors should address other possible mechanisms such gravitational settling. It could be argued that the initial massive injection included a variety of aerosol types and sizes some of which would have been removed by gravitational settling on time scales similar to that of photolysis.

4. We do consider gravitational settling in the manuscript, as well as several other explanations, but it could be a good idea to make our text more explicit. To further clarify we have added some text relating settling velocity to the extratropical downward transport large-scale circulation in the first paragraph of the Discussion section.

**Specific Comments**

A key step in the retrieval of aerosol properties from lidar observations is the selection of the lidar ratio. The choice of lidar ratio involves specific assumptions on the polydispersion particle size distribution, particle shape and complex refractive index. Although, not explicitly stated in the manuscript, one can assume that the authors considered the implicit lidar ratio assumptions (and the associated aerosol model) in the standard CALIOP level 2 product (Omar et al., 2009) to be inadequate for the interpretation of CALIOP observations in the presence of stratospheric carbonaceous aerosols and, therefore, decided to carry out their own inversion of CALIOP's Level 1b data making use of an improved aerosol model. A brief description of both the standard and adopted aerosol properties should be presented along with the rationale leading to the re-interpretation of CALIOP level1b data.

5. In part we have addressed this comment in answers 1 and 3 above. Here we would like to add that we did not chose a lidar ratio, instead we computed the lidar ratio in an iterative fitting procedure.

The authors discuss the evolution of the lidar, color and depolarization ratios in detail. Although the technical definitions of these terms are well known to the lidar community, the manuscript fails to connect the variability of those parameters with the variability on the actual microphysical and optical properties of the aerosol layer of great interest to readers beyond the lidar the community. A discussion of results in terms of practically meaningful aerosol properties will enhance the science value of this contribution.

6. We agree. We have added explanations in section 3.1 and 3.4.

The manuscript includes several statements on assumed numerical values of parameters without any references. No doubt most lidar experts are familiar with those values, but references may be important for the at large aerosol community. Please provide references and/or a rationale for the quoted numerical values in the statements below

Line 135. ....When the depolarization ratio is less than 0.05 the data is considered background and the lidar ratio is set to 50 sr.

7. OK, we added a reference to Vernier et al. (2009).

Line 137. Ice-clouds were removed in the lowest 3 km of the stratosphere by identifying them in stratospheric layers where the backscattering was high (attenuated backscattering larger than 0.0025 km-1 sr-1).

8. We have added a sentence that explains that we need to avoid statistical influence on cloud detection.

Line 138-139 Data in these layers were classified as probable clouds if their δv was higher than 0.20, or smoke if δv was between 0.05-0.20..

9. We have reformulated and clarified in the manuscript.

Line 164 …..very dense with layer AODs exceeding 1.

Detailed observations of the rapid evolution of the stratospheric AOD during the first two weeks following the onset of the pyroCb were carried out by the DSCOVR-EPIC mapper and AERONET ground-based observations (Torres et al., 2020). Both measurements reported stratospheric AOD's significantly larger than 1.0.

10. Thank you, this is a good suggestion. We have added text with reference to Torres et al., (2020).

Line 333. The UV aerosol index is available from a variety of UV-capable sensors on several platforms (Aura-OMI, SNPP-OMPS, DSCOVR-EPIC, S5P-TROPOMI). It should be pointed out that, when data on height of absorbing aerosol layers is available, the UVAI information content can be quantified in terms of the physically meaningful aerosol optical depth (AOD) and single scattering albedo (SSA) parameters. DSCOVR-EPIC near UV observations (Torres et al., 2020) were used to quantitatively (380 nm AOD and SSA) describe the 2017 Pacific Norwest pyroCb-triggered stratospheric aerosol layer on the first week following the injection using CALIPSO-provided aerosol layer height information.

11. Thank you, we have added that information in section 2.5.

Line 382. An objective evaluation of the accuracy of OMPS-LP -CALIOP measured aerosol extinction should be done using SAGE III as standard reference. Unlike CALIOP and OMPS LP aerosol extinction, SAGE III solar occultation observations require no aerosol model assumptions whatsoever. Although SAGE III measurements are spatially and temporally sparse, just a few collocations would be sufficient to assess the accuracy of the reported aerosol extinction products.

12. At first this suggestion has some appeal. However, in practice it is very difficult to compare SAGE III/ISS to the other two instruments. The reason is that the results of the two limb-oriented instruments report results in the tangent point, which is only one point in the line of sight. Since the lines of sight differ between the two instruments the results differ (Bourassa et al., 2019; JGR-Atmospheres, doi.org/10.1029/2019JD030607). Problem of not observing the same aerosol also affects a comparison with CALIOP. To overcome this obstacle a large amount of data from SAGE III/ISS would be needed to produce averages like those we formed for CALIOP and OMPS-LP. However, that is not possible because SAGE III/ISS produces such a small amount of data and is carried in a sub-optimal orbit for such a purpose, which we point out in the manuscript. See also our answer 3 concerning accuracy of the CALIOP results presented here.